# Concern about the Effectiveness of mRNA Vaccination Technology and Its Long-Term Safety: Potential Interference on miRNA Machinery

**DOI:** 10.3390/ijms24021404

**Published:** 2023-01-11

**Authors:** Gianmarco Stati, Paolo Amerio, Mario Nubile, Silvia Sancilio, Francesco Rossi, Roberta Di Pietro

**Affiliations:** 1Department of Medicine and Ageing Sciences, G. d’Annunzio University of Chieti-Pescara, Via dei Vestini, 31, 66100 Chieti, Italy; 2Ophthalmology Clinic, National Centre of High Technology (CNAT) in Ophthalmology, G. d’Annunzio University of Chieti-Pescara, Via dei Vestini, 31, 66100 Chieti, Italy; 3Department of Molecular Sciences and Nanosystems, Ca’ Foscari University, Via Torino, 155/b, 30170 Venice, Italy; 4Sbarro Institute for Cancer Research and Molecular Medicine, Center for Biotechnology, Department of Biology, College of Science and Technology, Temple University, Philadelphia, PA 19122, USA

**Keywords:** COVID-19, mRNA vaccines, miRNAs, SARS-CoV-2, innovative formulation, cancer, mRNA1273, BNT162b2

## Abstract

After the outbreak of the pandemic due to COVID-19 infection, several vaccines were developed on short timelines to counteract the public health crisis. To allow the administration of mRNA vaccines through a faster-paced approval process, the Emergency Use Authorization (EUA) was applied. The Ba.5 (omicron) variant of SARS-CoV-2 is the predominant one at this moment. Its highly mutable single-stranded RNA genome, along with its high transmissivity, generated concern about the effectiveness of vaccination. The interaction between the vaccine and the host cell is finely regulated by miRNA machinery, a complex network that oversees a wide range of biological processes. The dysregulation of miRNA machinery has been associated with the development of clinical complications during COVID-19 infection and, moreover, to several human pathologies, among which is cancer disease. Now that in some areas, four doses of mRNA vaccine have been administered, it is natural to wonder about its effectiveness and long-term safety.

## 1. Introduction

Since the outbreak of the pandemic in 2020, significant efforts have been made to develop new vaccines to counteract COVID-19 infection and reduce mortality and morbidity among infected people [1]. Under the guidelines established by the Centers for Disease Control and Prevention (CDC) in the United States, the normal process for developing a vaccine and the completion of all of the phases would take about 10 years. It has been reported by the Center for Biologics Evaluation and Research, a division of the US Food and Drug Administration (FDA) that oversees vaccine regulation. To face the urgent public health crisis, the FDA recognized the need to bypass many of the normal restrictions to speed up vaccine development. This is the case of the mRNA vaccines, a completely new vaccine technology that has never been used before. The principal means to counteract the global crisis caused by the coronavirus pandemic was to apply for processing EUA to allow these treatments to be administrated in a state of emergency [2]. Around the world, the percentage of short-term adverse reactions has been shown to be limited in number [3], since only transient local, and systemic reactions in the short- and medium-term have been assessed [4]. Now that millions of recipients are receiving doses of mRNA vaccines, it is natural to wonder about the long-term safety of such vaccines.

## 2. Different SARS-CoV-2 Variant of Concern and Effectiveness of Vaccination

Nowadays, the link between susceptibility to COVID-19 and the specific characteristics of individual patients is not yet understood. SARS-CoV-2 exhibits a highly mutable single-stranded RNA genome, which is the reason why several variants of the SARS-CoV-2 are growing worldwide, causing multiple waves of outbreaks of this viral illness. Nevertheless, the acquisition of mutations due to SARS-CoV-2 spreading failed to explain regional diffusion heterogeneity [5].

Omicron has five lineages, Ba.1 (B.1.1.529.1), Ba.2 (B.1.1.529.2), Ba.3 (B.1.1.529.3), Ba.4, and Ba.5, which were first detected in November 2021 in South Africa [6]. The Ba.5 variant of SARS-CoV-2 is the predominant one at this moment. New Omicron variants have recently appeared, containing identical receptor-binding domain (RBD) sequences to Ba.2 but with the addition of L452R and F486V substitutions, respectively, named Ba.2.12.1 and Ba.2.13. All displayed higher transmission advantages over Ba.2 [7]. 

The exchange of a single amino acid can drastically affect a virus’s ability to evade the immune system and nullify the progress of vaccine development against the virus. In fact, SARS-CoV-2, similar to other RNA viruses, is prone to genetic evolution while adapting to its new human hosts with the potential development of new mutations, resulting in the emergence of multiple variants compared to its ancestral strains [8].

Data from late December 2021 and early January 2022 indicate that Omicron Ba.2 is about 1.5 and 4.2 times as contagious as Ba.1 and Delta, respectively [9]. Omicron evolved mutations to evade the humoral immunity elicited by Ba.1 infection, suggesting that the Ba.1-derived vaccine may not achieve broad-spectrum protection against the new Omicron variants [10]. The European Centre for Disease Prevention and Control (ECDC) observed an increase in the proportion of Ba.4 and Ba.5 infections in many EU/EEA countries [11], reclassifying Omicron sub-lineages Ba.4 and Ba.5 from variants of interest to variants of concern [12]. The prevalence of the Ba.4 and Ba.5 variants of concern, compared to the ancestral variant Ba.2, is probably due to their ability to evade immune protection induced by prior vaccination or infection, which also tends to wane over time rapidly [13].

The very last emerged BA.2.75 SARS-CoV-2 variant, also known as ‘Centaurus’, exhibits nine additional S mutations compared to the BA.2 variant and enhanced neutralizing antibody escape in mRNA-vaccine recipients [14]. Moreover, the BA.2.75 variant shows a strengthened cell–cell fusion, driven mainly by the N460K mutation. It allows new receptor contact, supporting a mechanism of potentiated syncytia formation [15]. It is thought that Omicron Ba.2.75 is on its path to becoming the next dominating variant all over the world. The high transmissivity of the newest variants generated concern about the effectiveness of vaccination and its long-term safety now that it has even reached the fourth dose in some areas of the world [16].

## 3. Immune Response and Circulating Extracellular Vesicles Containing miRNAs

Recent studies showed that circulating extracellular vesicles containing microRNAs are strongly associated with antibody production and adverse reactions after COVID-19 vaccination [17]. The administration of the mRNA vaccines, known as mRNA-1273 and BNT162b2, respectively, is proved to trigger a whole series of biomolecular reactions within the cell [18], where single strands of viral mRNA are allowed to freely circulate once they have penetrated the cell by endocytosis. MiRNAs are non-coding RNA molecules made of 20–24 nucleotides that regulate post-transcriptional gene expression.

MiRNAs have emerged as regulators of the immunity-related gene targets through complex networks of vaccine–host cell interactions, as they have actions on a wide range of biological processes. They can act as negative post-transcriptional regulators of gene expression in the cytoplasm due to strong complementarity to specific mRNA targets [19]. MiRNAs are epigenetic regulators of gene expression patterns, and they modulate co-transcriptional alternative splicing events [20]. Numerous miRNAs appear to be critically implicated in cell differentiation as well as in the maintenance of differentiated states [21].

The scientific evidence indicates that the genes encoding for miRNAs have complex transcriptional patterns that respond to various stimuli, such as cellular signaling pathways, including biotic stress, as well as decay mechanisms [22]. Several lines of evidence demonstrate that miRNAs participate in essential mechanisms of cell biology, the regulation of the immune system, and the onset and progression of numerous types of disorders.

Along with circulating extracellular vesicles containing microRNAs, a pivotal role of the S1 spike protein in COVID-19 inflammation pathogenesis was described. S1 spike protein might be not only responsible for the adherence and the invasion of the virus to the host cells and the onset of the cytokine storm but also responsible for long-COVID syndrome [23]. Moreover, it was estimated that a greater number of free S1 spikes in the blood during COVID-19 infection is correlated with a poor prognosis of COVID-19 [24]. Fortunately, the huge number of the accumulated mutation in the spike protein of the latest variants of concern results in a disrupted recognition of certain Toll-like receptors (TLR), resulting in the lower activation of the NF-κB pathway and related signaling pathways [25].

## 4. Known Adverse Reactions of mRNA Vaccines

The intramuscular administration of BNT162b2 and mRNA-1273 vaccines has been proven to generate a robust humoral immune response against COVID-19 while causing a relatively limited number of adverse reactions. Rosenblum et al., 2022, reported an observational study of reports to the Vaccine Adverse Event Reporting System (VAERS), one of the main control bodies in place to monitor the safety of the vaccines, where the administration of 298,792,852 doses in the US have led to 340,522 adverse reactions. Of these reports, 22,527 were deemed serious, while 4496 caused the death of the patients (Figure 1A) [26].

The majority of the recipient of mRNA vaccinations experience minor side effects, such as pain at the injection site or fatigue, headache, and myalgia [27]. Serious adverse reactions were reported to be in excess of 15 cases in 10,000 patients for mRNA-1273 and 10.1 cases in 10,000 for BNT162b2 when compared with the placebo control groups (6.4 in 10,000 subjects and 2.3 in 10,000 subjects, respectively, Figure 1B) [28].

Serious adverse reactions were more common after the injection of the second dose and included: myocardial infarction [29], myocarditis/pericarditis [30], cerebral venous sinus thrombosis [31,32], and hypermetabolic lymphadenopathy [33].

Myocarditis and myocardial infarction were reported by multicenter statistical analysis both in Israel and the US [29,30], with an incidence of 1/20,000 for male patients in the age range between 16 to 30 years and a general incidence between all the patients of 1/100,000 against an expected average incidence of 0.12–0.15/100,000 cases for male patients and of 0.07–0.08/100,000 cases for female patients. A prevalence of cases in male patients of every age group and a prevalence of myocarditis in the range of 12–24 years, regardless of sex, has been highlighted [34].

Cerebral venous sinus thrombosis has been reported to be a symptom of SARS-CoV-2 and an adverse reaction of the virus-inactivated vaccines [31,32], but a lower rate has also been registered after the administration of mRNA vaccines, representing 0.07% of all serious adverse reactions reported in the World Health Organization pharmacovigilance database (756 cases by the 30 September 2021) [35]. Of the two mRNA vaccines investigated in this work, BNT162b2 caused the majority of the cases, 620 (0.06%), while 136 (0.01%) cases were attributed to mRNA-1273, with a predominance of cases among female patients [35].

Cohen et al., 2021, considered a total of 728 oncologic patients that underwent PET-CT studies in order to differentiate between the malignant hypermetabolic lymphadenopathy (MHL) and the benign nature of the vaccine-associated hypermetabolic lymphadenopathy (VAHL) after mRNA vaccination [36]. Hypermetabolic lymphadenopathy (HLN) has been reported in the 45.6% of fully vaccinated recipients, regardless of previous conditions [36], with various degrees of intensity. Of these reports, 80.1% were attributed to the effect of the vaccination, and it was resolved in the 5 days following the inoculation [37].

Other adverse reactions, such as Bell’s palsy, strokes, appendicitis, and autoimmunity, have been reported to be connected with mRNA vaccinations [38], but there is no clear evidence or consensus between the research groups that mRNA vaccinations significantly change the occurrence of these diseases.

## 5. Discussion

The disruption of miRNA biogenesis machinery is responsible for several human pathologies [39]. miRNA dysregulation is associated with the development of clinical complications during COVID-19 infection [40]. It has been demonstrated that the expression level of numerous miRNAs is altered after COVID-19 vaccination [41]. Thus, the altered expression levels of circulating miRNAs could affect the severity of the disease once contracted. SARS-CoV-2-encoded miRNAs can affect the host’s immune response [42]. It is reasonable to assume that the disrupted expression of these small molecules can contribute to the onset of other longer-term diseases. The dysregulation of the host miRNA range that modulates multiple gene expressions can influence cancer development, either directly or indirectly [43]. In fact, different miRNAs can function as oncogenes or tumor-suppressor genes [44]. For instance, recent studies have revealed that miRNA-451 (miR-451), downregulated after COVID-19 vaccination, is involved in various human physiological and pathological processes [45]. It has been shown that miR-451 is implicated in the progression of multiple cancer types, affecting the tumor cells’ invasiveness and metastasis after secretion via exosomes into the tumor microenvironment, not only directly but also indirectly.

Based on vaccination schedules [46], any single dose contains a huge number of viral mRNAs, resulting in their massive entry into the host cells. Comparable to a ‘Trojan horse’, it can be assumed that a whole series of metabolic events will be triggered as a positive feedback loop in any given gene network that is regulated by miRNAs. It is also known that most of the miRNAs that operate in the nucleus will simultaneously regulate transcript stability in the cytoplasm and vice versa, resulting in a highly integrated mechanism. As cellular metabolic pathways follow the law of mass action, a substantial number of miRNAs targeted to numerous viral mRNAs should be produced. It is also possible that the host’s miRNA machinery might be overwhelmed in the processing of miRNAs and diverted away from its normal cellular functions and molecular pathways. This might lead to dangerous, long-lasting dysregulation of the miRNA pathways. It is reasonable to assume that some viral messenger RNAs will trigger changes in the host miRNA transcription profiles or stabilities and that the resulting modified miRNA clusters might favor the development of different disorders. This might be particularly relevant if the altered miRNA cistron included host miRNAs with oncogenic properties.

## 6. Conclusions

As part of a World Health Organization (WHO) ad hoc consultation on the ongoing evaluation of the safety of COVID-19 mRNA vaccines, a general consensus has emerged. Nowadays, COVID-19 vaccines are safe and effective. They were evaluated across tens of thousands of participants in clinical trials. The mRNA vaccines met the Food and Drug Administration’s (FDA’s) rigorous scientific standards for safety, effectiveness, and manufacturing quality. Pfizer-BioNTech and Moderna will continue to undergo the most intensive safety monitoring in history. This monitoring includes using both established and new safety monitoring systems to ensure that COVID-19 vaccines are safe. As is well known, some people have side effects after receiving their COVID-19 vaccine, while others might have no side effects. Short-term side effects may affect the ability to perform daily activities, going away within a few days. In rare cases, people have experienced serious health events after COVID-19 vaccination. Any health problem that happens after vaccination is considered an adverse event that can be caused by a coincidental event not related to the vaccine itself, such as an unrelated fever that happens following vaccination. VAERS revealed rare and serious types of adverse events following COVID-19 vaccination, although rare, along with rare cases of death, but it is unclear whether the vaccine was the cause. Despite the promising positive short-term outcomes, it is ethically appropriate to continue the blinded follow-up of placebo recipients in existing trials to further define the longer-term safety of these mRNA vaccines. This type of technology would appear to be very promising, and if the expectations were confirmed, it might be adopted in many other drug formulations.

## Figures and Tables

**Figure 1 ijms-24-01404-f001:**
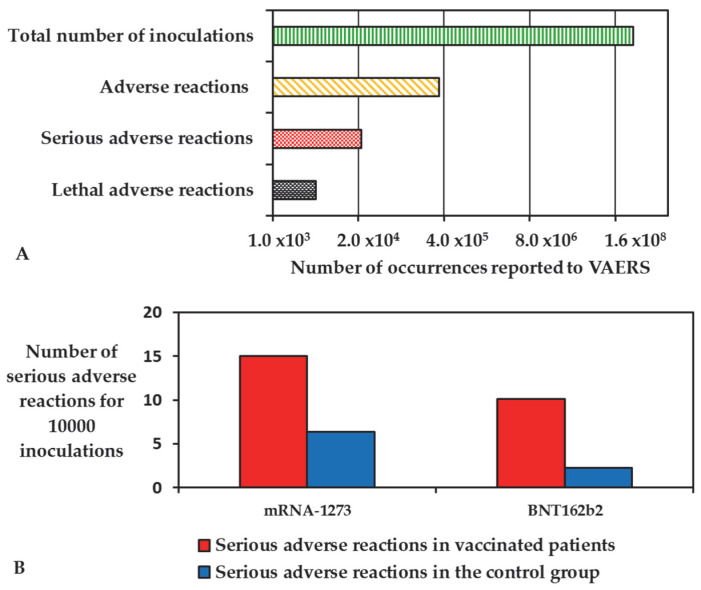
(**A**) Adverse reactions reported to the VAERS database in comparison with the total inoculations in US after 6 months of vaccination campaign; (**B**) comparison between the serious adverse reactions reported during the clinical trials of the mRNA vaccines and their placebo control groups.

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
