# Peer review of "Concern about the Effectiveness of mRNA Vaccination Technology and Its Long-Term Safety: Potential Interference on miRNA Machinery"

_ijms, 2023, doi:10.3390/ijms24021404_

Round 1

Reviewer 1 Report

The manuscript by Stati et al "Concern about the effectiveness of mRNA vaccination technology and its long-term safety" addresses an important topic related to mRNA vaccination technology in infectious diseases but also for other indications, such as cancer. The manuscript is written clearly,  however, it refers to the data regarding efficacy and safety rather superficially, without a comprehensive review of existing data, which might be attributed to the article type, i.e. Opinion letter.

The manuscript focusses on ONE interesting aspect of mRNA vaccination, i.e. the interaction with the miRNA machinery of the cell and addresses (also refering to respective references) their potential correlation to efficacy on the one hand and clinical complications on the other.

While the topic of effects on the miRNA machinery is certainly an interesting and relevant, the author do not look at sufficient degree at additional / alternative mechanisms affected by mRNA vaccination, such as the potential role of the encoded Antigen, i.e. Spike protein with multiple effects, including activation of TLR and proinflammatory cellular pathways (e.g. NFkappaB, etc. and effects on the coagulation system. The authors are encouraged to provide some more details and references to interactions beside miRNA disturbances.

Since a more comprehensive covering of all relevant aspects may be beyond the format of an Opinion letter, the authors may chose to narrow the title / topic of the article more specifically to miRNA interferences.

Reviewer 2 Report

The authors provide good summary of the mRNA vaccine efficacy, side-effect, and some molecular mechanisms behind it. As this novel mRNA plays critical role in this pandemic and may be an important tool for people to antagonize pathogens. It is of interest to provide summary and opinions on this regard. However, when the authors discussed the side effect of the vaccines, they did not provide data for control groups, which is essential for the readers to understand the real case number caused by the vaccines themselves. Moreover, a table or bar graph would make it easier to interpret the data. I listed my specific concerns below.

1. Line 112. How about the control group? Please calculate the percentage of death for each group. Please calculate the percentage and summarize in a table or bar graph.

2. Line 117. How many cases in control group? Please calculate the percentage and summarize in a table or bar graph.

3. Line 118. Incidence?

4. Line 124. What about the control group?

5. Line 126. It would be a symptom rather than an adverse reaction of the disease COVID-19, What is the total incidence? What about control group? And COVID-19 is a disease caused by SARS-CoV-2 infection. The narrative should be revised.

6. Line 132. What about the control?

Round 2

Reviewer 1 Report

The revised manuscript can be recommended for publication  

Author Response

Thanks to the comments of the Reviewer 1 the manuscript has been definitely improved.

Reviewer 2 Report

The authors have significantly addressed my concern. My last comment is that whenever a control group is not available, this should be clearly stated and discussed.

Author Response

REVIEWER 2

The authors have significantly addressed my concern. My last comment is that whenever a control group is not available, this should be clearly stated and discussed.

The authors would like to thank the Reviewer 2 for its recommendation. The text was improved according to the Reviewer comment.